# Analysis of the temporal and spatial evolution of China's air passenger transportation and the impact of policies on passenger flows

Chen Luo *, Tianshun Ma, Min Wang, Xinrong Hu, Chunyang Liu, Xutong Wang

Air Traffic Management Institute, Civil Aviation Flight University of China, Guanghan, Sichuan Province, China

* luochenkun@cafuc.edu.cn

## Abstract

The spatial-temporal evolution of air passenger flows in China's aviation sector is critical for optimizing route networks and capacity allocation, particularly in assessing the efficacy of policy interventions to inform future regulatory frameworks. This study leverages air passenger throughput data spanning prefecture-level cities (2005–2023) and employs the Standard Deviation Ellipse (SDE) method to identify spatial agglomeration patterns in 2008, 2017, 2020, and 2022. A Difference-in-Differences (DID) model is further applied to quantify the causal effects of policy interventions on air transportation dynamics, supplemented by robustness tests and heterogeneity analyses to validate the reliability of the findings. Three key conclusions emerge from the analysis. First, while the spatial distribution of air passenger traffic predominantly aligns with a northeast-southwest axis, the strengthened north-south directional influence (reflected in the increased major-to-minor axis ratio of the SDE) indicates a bidirectional expansion of air transport coverage. Second, the spatial centroid of passenger flows exhibited a directional shift from northwestern to southwestern China between 2005 and 2023, highlighting the rising strategic importance of the southwestern region. Notably, by 2023, a reversed spatial transition emerged, with the centroid gradually reverting toward the northeast, signaling a recalibration of passenger flow patterns. Third, policy impacts display marked regional and urban heterogeneity: eastern China experienced significantly higher passenger growth compared to central, western, and northeastern regions, while provincial capitals and megacities—owing to their resource concentration and policy implementation capacity—demonstrated stronger responsiveness to regulatory measures than smaller prefecture-level cities. These findings provide actionable insights for both industry and policymakers. Airlines may utilize the empirical evidence to refine route planning and capacity adjustments, whereas policymakers could prioritize spatially differentiated regulations to mitigate regional imbalances and foster integrated national aviation development.

**Data availability statement:** Data publicly deposited in DRYAD, DOI: 10.5061/dryad.83bk3jb33.

**Funding:** This manuscript was funded by the Tibet Autonomous Region Science and Technology Program (Project XZ202403ZY0014 to C.L.).

**Competing interests:** The authors have declared that no competing interests exist.

## 1. Introduction

Since the deregulation of the aviation industry in the 1980s, China's civil aviation sector has experienced exponential growth, gradually becoming the world's second-largest aviation market [1]. Air transportation not only functions as a critical modality for intercity mobility but also functions as a key link in spatial connectivity, reflecting the evolution of urban transportation networks to a certain extent [2]. Policy interventions exert a substantial influence on the spatiotemporal evolution of the aviation transport system, making it an exemplary case for understanding the evolution of air transport systems and the impact of policies. China's strategic infrastructure investments and regulatory reforms have driven an average annual passenger growth rate surpassing 10% [3,4]. Key policy measures include elevating on-time performance thresholds to 85% for operational efficiency improvement [5], incentivizing new airline entrants [6], and executing airport network expansion plans [7].These policies have reshaped air traffic patterns by reducing the dominance of major hub airports in national passenger traffic while promoting air passenger growth in western regions. Furthermore, timely regulatory interventions in response to emergencies have facilitated the rapid recovery of China's aviation industry [8], underscoring the role of policies in shaping the spatial structure of air transport [9].

Research on aviation transport networks has extensively examined their structural evolution, spatial distribution patterns, and underlying determinants of passenger flows. A prominent focus has been the transformation of China's aviation network, which has shifted from a monocentric structure dominated by Beijing to a more decentralized, diamond-shaped framework with multiple core hubs, including Beijing, Shanghai, Guangzhou, and Chengdu [10]. This transition reflects broader trends toward spatial diversification in the aviation market, driven by factors such as airport hierarchy, the rise of low-cost carriers, and competitive pressures from high-speed rail expansion [11].Complex network analyses further reveal that the density of air routes and passenger traffic exhibit a strong positive correlation, suggesting that connectivity reinforces demand [12]. Studies employing network metrics and geographically weighted regression models highlight the "small-world" properties of China's aviation system, where hub airports play a disproportionately influential role in maintaining network efficiency and connectivity [13,14]. These findings underscore the dual importance of spatial distribution and network topology in shaping the performance and evolution of aviation systems.

The spatial distribution of airports plays a critical role in shaping air passenger flows and market dynamics. Recent studies have employed spatial econometric models to examine the clustering patterns of Chinese airports, revealing strong regional interdependencies in their development [15]). Notably, China's aviation infrastructure exhibits significant geographical concentration, with major hubs predominantly located in the central and eastern regions, forming high-density flight networks around Beijing, Shanghai, Guangzhou, and the Chengdu-Chongqing corridor [16]. Further research has explored the emergence of multi-airport regions (MARs), integrating high-speed rail connectivity as a key factor in optimizing airport layouts [17].

Beyond spatial analysis, spectral clustering techniques have been applied to classify airports based on their functional roles within the broader aviation network, identifying critical nodes that drive connectivity [18].

The interplay between economic factors and air passenger flows has also been extensively investigated. Geographically weighted correlation analyses demonstrate substantial spatial heterogeneity in the relationship between regional economic performance and aviation activity [19]. Hierarchical structures in air transport have been examined through Zipf's law and geographical detector models, highlighting the influence of international trade, foreign investment, and business travel on market formation [20]. Additional studies emphasize the dual impact of GDP growth and tourism on aviation networks, alongside competitive and complementary effects introduced by high-speed rail expansion [21]. Temporal analyses further reveal lagged interactions between air transport and economic development, particularly in regions like the Yangtze River Delta, where global and domestic economic fluctuations exert delayed influences [22].

To assess spatial evolution in air transport, the standard deviation ellipse (SDE) method has been widely adopted since its introduction by Lefever (1926). Applications of this approach reveal divergent growth patterns across Chinese provinces, with Hebei experiencing rapid per capita passenger throughput growth, while Beijing's growth rates lagged behind. Western and southeastern coastal regions, meanwhile, emerged as leaders in per capita throughput [23]. Longitudinal studies tracking airport distribution from 2000 to 2019 identify a northwestward expansion of air transport coverage, reflecting broader regional development trends [24]. Similarly, analyses of passenger and cargo throughput demonstrate a gradual shift toward more balanced demand distribution across China's aviation network [25].

In causal inference research, the difference-in-differences (DID) method has become instrumental in evaluating policy impacts on air transport. Aviation studies leveraging DID models have uncovered heterogeneous regional effects of airport construction and high-speed rail competition, underscoring the method's versatility in isolating causal relationships [26,27]. Parallel applications in other transport sectors—such as railways, roads, and maritime shipping—further validate the method's robustness. For instance, high-speed rail operations were found to disproportionately benefit core cities at the expense of peripheral service industries [28], while simultaneously stimulating regional innovation and economic growth [29,30]. Road transport studies highlight the uneven outcomes of expressway expansions on poverty alleviation and regional economies [31,32], whereas maritime research confirms the efficacy of port policies in adapting to external shocks like canal expansions [33,34]. Within aviation, DID models have quantified the environmental impact of airport operations [35], market competition dynamics following new airline entries [36], and the substitution effects of high-speed rail on short-haul flights [37].

Although existing research extensively explores aviation transport networks, airport spatial distribution, and economic factors, studies on the impact of policies on air passenger flows remain relatively limited. Furthermore, most studies focus on short-term policy effects, with insufficient systematic analysis of the cumulative impact of long-term policy interventions, the lagged adaptation of aviation networks to policies, and policy effects in the context of unexpected events. Therefore, this study employs the DID method to quantify the causal effects of policy interventions on air passenger flows, filling gaps in existing research while further validating the applicability of the method in aviation transport studies. Specifically, this study first utilizes the SDE method to analyze the spatiotemporal evolution of China's air passenger transport from 2005 to 2023 and subsequently applies the DID method at the prefecture-level city scale to quantify the impact of passenger flow guidance policies on air transport. This research not only enhances the understanding of policy-driven changes in the aviation market but also provides a scientific basis for policy formulation.

## 2. Data sources and research methods

### 2.1. Research objects

The study encompasses the following administrative divisions: 22 provinces (Hebei, Shanxi, Liaoning, Jilin, Heilongjiang, Jiangsu, Zhejiang, Anhui, Fujian, Jiangxi, Shandong, Henan, Hubei, Hunan, Guangdong, Hainan, Sichuan, Guizhou,

 

Yunnan, Shaanxi, Gansu, Qinghai, and Taiwan), 5 autonomous regions (Inner Mongolia, Guangxi, Tibet, Ningxia, and Xinjiang), and 4 municipalities directly under the central government (Beijing, Tianjin, Shanghai, and Chongqing).Notably, the Hong Kong Special Administrative Region and Macao Special Administrative Regions, along with the Taiwan region, are excluded from the analysis due to data availability constraints.

## 2.2. Research data

Airport operational metrics and passenger throughput data were synthesized from the National Civil Aviation Airport Production Statistical Bulletin (2005–2023).Socioeconomic indicators, including regional GDP and tourism revenue, were extracted from provincial-level Statistical Yearbooks, National Economic and Social Development Statistical Bulletins, and official portals of provincial Culture and Tourism Bureaus [38]. Only prefecture-level cities hosting airports were included in the analysis, with cities exhibiting significant missing data being excluded.Spatial visualization was implemented using georeferenced cartographic layers from the Baidu Maps API (Application Programming Interface).

## 2.3. Research methods

### 2.3.1. Standard deviation ellipse.
The SDE method [39] is an effective geostatistical analytical framework that characterizes the spatial distribution of a research object from the perspectives of centrality, spread, and directionality. Extensively employed in spatial economics research, the SDE method enables scholars to reveal the multidimensional characteristics of economic spaces and their dynamic changes. In this study, the standard deviation ellipse is applied to analyze the evolution of the spatial distribution pattern of air passenger traffic across China's airports, thereby enabling a systematic deconstruction of the changes and development trends in air passenger traffic from 2005 to 2023.

$$\bar{X}_w = \frac{\sum_{i=1}^{n} w_i x_i}{\sum_{i=1}^{n} w_i}, \bar{Y}_w = \frac{\sum_{i=1}^{n} w_i y_i}{\sum_{i=1}^{n} w_i} \tag{1}$$

$$S = \pi \sigma_x \sigma_y \tag{2}$$

$$R = \frac{\sigma_x}{\sigma_y} \tag{3}$$

$$\tan\theta = \frac{\left(\sum_{i=1}^{n} w_i^2 \bar{x}_i^2 - \sum_{i=1}^{n} w_i^2 \bar{y}_i^2\right) + \sqrt{\left(\sum_{i=1}^{n} w_i^2 \bar{x}_i^2 - \sum_{i=1}^{n} w_j^2 \bar{y}_i^2\right)^2 + 4\sum_{i=1}^{n} w_i^2 \bar{x}_i^2 \bar{y}_i^2}}{2\sum_{i=1}^{n} w_i^2 \bar{x}_i^2 \bar{y}_i^2 {}_i} \tag{4}$$

$$\sigma_x = \sqrt{\frac{\sum_{i=1}^{m} \left(w_i \bar{x}_i \cos\theta - w_i \bar{y}_i \sin\theta\right)^2}{\sum_{i=1}^{n} w_i^2}}, \sigma_y = \sqrt{\frac{\sum_{i=1}^{m} \left(w_i \bar{x}_i \sin\theta - w_i \bar{y}_i \cos\theta\right)^2}{\sum_{i=1}^{n} w_i^2}} \tag{5}$$

To systematically deconstruct the spatiotemporal distribution patterns of air passenger throughput, this study employs the Standard Deviation Ellipse (SDE) method to quantitatively describe the spatial and temporal distribution changes of the air transportation network. Specific parameters include: the weighted centroid $(\bar{X}_w, \bar{Y}_w)$ represents the spatial mean center of air passenger throughput distribution of the air passenger throughput distribution; $(w_i)$ represents the weight of

the air passenger throughput at (i); $(x_i, y_i)$ represents the spatial coordinates of the air passenger throughput; the area of the ellipse ($S$) is used to describe the spatial coverage of the air passenger throughput; ($\pi$) is a constant, and the value is taken to be 3.14; and the standard deviation ellipse long and short axes ($\sigma_x$) and ($\sigma_y$) indicate the primary and secondary distribution direction trends of the passenger throughput, respectively. throughput trends in the primary and secondary distribution directions, respectively. The long axis reflects the main extension direction of passenger activity, while the short axis indicates the distribution of the secondary extension direction; the ratio of the two axes of the ellipse ($R$) is used to characterize the spatial isotropy of the distribution of passenger throughput; the azimuth angle ($\theta$) indicates the main directional tendency of the air passenger activity, i.e., the preference of the passenger throughput along a particular direction; ($\bar{x}_i$) and ($\bar{y}_i$) represent the weighted mean center of the air passenger throughput.

Through parametric decomposition of SDE outputs, this research quantifies the spatiotemporal polarization and directional bias inherent in China's aviation network evolution. This spatial distribution analysis, based on the standard deviation ellipse method, helps to visualize the impact of various policies on the air transport network.

**2.3.2. Double difference modeling.** The DID method [40] has been widely applied in policy evaluation, and this paper uses the DID model to assess the changes in the impact of passenger flow guidance policies on passenger traffic through empirical research. The main advantage of the DID model is its ability to exploit policy exogeneity, address endogeneity bias, and avoid reverse causality issues. Additionally, the fixed-effects estimation in the DID model helps reduce bias caused by omitted variables [41].

The calculation formula for the DID model is as follows:

$$P_{i,t} = \alpha_0 + \alpha_1 T_{i,t} + \alpha_2 Control_{i,t} + \mu_i + \eta_t + \varepsilon_{i,t} \tag{6}$$

Where: $i, t$ denotes city and year, respectively; $P_{i,t}$ is the explanatory variable; $\alpha_0$ is a constant term; $T_{i,t}$ is the core explanatory variable, $\alpha_1$ denotes the effect of landing and takeoff times on air passenger traffic; $Control_{i,t}$ is the control variable, $\alpha_2$ is the vector of coefficients for the control variable; $\mu_i$ is an individual fixed effect, $\eta_t$ is a time fixed effect; and $\varepsilon_{i,t}$ is the error term.

## 3. Evolution of the national air passenger transportation pattern

Based on China's air passenger throughput data for the period 2005–2023, the spatial distribution is visualized using ArcGIS. The SDE method is employed to delineate the movement of the center of gravity of the air passenger flow, as shown in Fig 1. In the following analysis, a five-year time frame is used. This temporal segmentation aligns with China's Five-Year Plan (FYP) institutional framework and social development model, which is profoundly shaped by the FYP. Since 1953, the Chinese government has set economic growth targets, guided industrial development, and adjusted social policies by formulating and implementing a national-level FYP every five years. Therefore, the FYP serves as a pivotal temporal metric in China's development process, effectively encapsulates policy-driven developmental trajectories of the country's progress at different stages.

### 3.1. Characterization of the evolution of the national passenger throughput pattern, 2005–2009

Between 2005 and 2009, China experienced a period of rapid economic growth, marking an economic acceleration phase. During this time, air passenger transportation demand experienced a steady increase, driven by the growing economy, expanding foreign trade, and greater globalization, all of which contributed to a surge in air passenger traffic.

As illustrated in Table 1 and Fig 2, the spatial centroid exhibited a northward migration during this period.

Between 2005 and 2009, the center of gravity of air passenger throughput shifted from 114°16′00″E, 31°18′23″N to 114°10′35″E, 31°33′25″N. This northward movement of the center of gravity accelerated, indicating that the northern region's passenger throughput was increasing at a faster pace. The speed of this northward shift underscoring the rapid expansion of the civil aviation market in northern China.

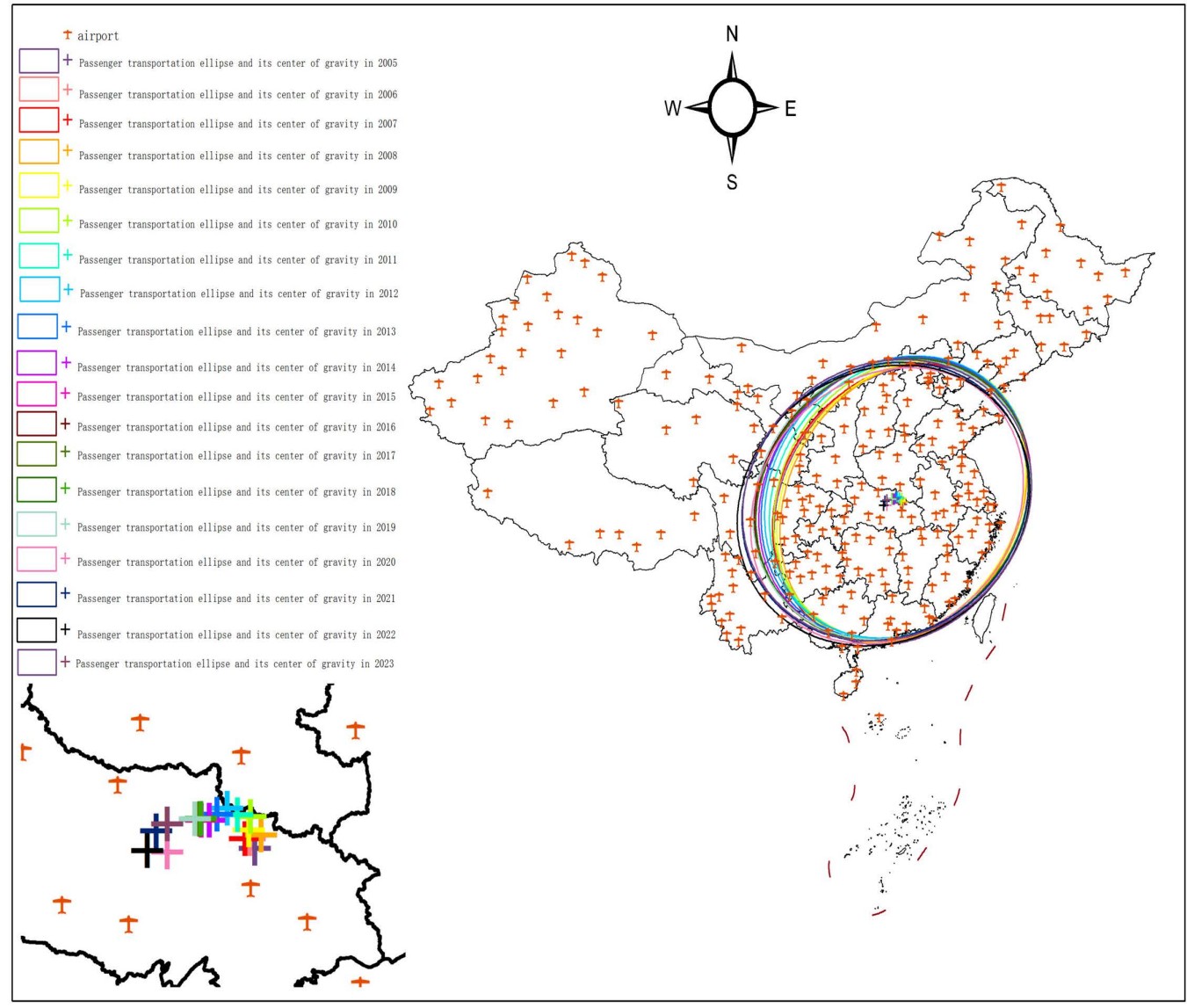

**Fig 1. National passenger throughput standard deviation ellipse and its center of gravity, 2005-2023.** The basic map came from the official web-site of Standard map of China at http://bzdtchinnr.gov.cn/. The drawing approval number is GS (2022) 4313. The data used was calculated by the author.

The concentration of passenger throughput showed a trend of first concentrating, then dispersing. In 2005, the standard deviation ellipse area of passenger throughput was 310.44 km², which decreased by 11.42 km² by 2008, suggesting a trend toward more centralized distribution. However, by 2009, the area had slightly increased to 300.13 km², reflecting a trend toward dispersion in the distribution of passenger throughput.

Throughout the 2005–2009 period, the standard deviation ellipse generally shifted westward. Notably, in 2008, it showed an aggregation effect and exhibited an eastward displacement. This shift was largely influenced by major events such as the Beijing Olympic Games and the Wenchuan earthquake. The Olympic Games drew significant numbers of tourists, athletes, and media, leading to a sharp increase in transportation demand. In response, China accelerated the construction and improvement of transportation infrastructure to accommodate the demand during the Games [42,43].

**Table 1. National passenger throughput standard deviation ellipse parameters.**

| Year | Area (km²) | Longitude | Latitude | Short axis (km) | Long axis (km) | Ratio of two axes | Azimuth (°) |
|------|-----------|-----------|----------|-----------------|----------------|-------------------|-------------|
| 2005 | 310.44 | 114°16′0″ | 31°18′23″ | 8.94 | 11.04 | 80.97 | 60.62 |
| 2006 | 308.16 | 114°11′40″ | 31°26′12″ | 8.86 | 11.06 | 80.10 | 60.34 |
| 2007 | 308.46 | 114°7′22″ | 31°26′22″ | 8.89 | 11.03 | 80.59 | 60.99 |
| 2008 | 299.02 | 114°21′33″ | 31°29′8″ | 8.73 | 10.89 | 80.16 | 53.65 |
| 2009 | 300.13 | 114°10′35″ | 31°33′25″ | 8.70 | 10.97 | 79.63 | 54.70 |
| 2010 | 310.90 | 114°12′18″ | 31°43′57″ | 8.98 | 11.00 | 81.63 | 60.70 |
| 2011 | 317.99 | 114°0′58″ | 31°45′28″ | 9.13 | 11.08 | 82.40 | 63.55 |
| 2012 | 327.04 | 113°51′48″ | 31°50′40″ | 9.26 | 11.24 | 82.38 | 66.99 |
| 2013 | 332.58 | 113°43′1″ | 31°45′43″ | 9.30 | 11.38 | 81.72 | 67.82 |
| 2014 | 333.99 | 113°36′16″ | 31°40′48″ | 9.27 | 11.45 | 80.96 | 68.22 |
| 2015 | 340.43 | 113°29′22″ | 31°41′3″ | 9.31 | 11.62 | 80.12 | 70.50 |
| 2016 | 338.90 | 113°26′29″ | 31°42′10″ | 9.27 | 11.63 | 79.70 | 70.53 |
| 2017 | 335.89 | 113°28′0″ | 31°41′34″ | 9.24 | 11.56 | 79.93 | 68.77 |
| 2018 | 336.23 | 113°28′53″ | 31°42′20″ | 9.25 | 11.56 | 80.01 | 69.73 |
| 2019 | 338.79 | 113°23′9″ | 31°40′20″ | 9.26 | 11.63 | 79.62 | 72.69 |
| 2020 | 327.01 | 112°59′51″ | 31°15′38″ | 9.01 | 11.54 | 78.07 | 68.85 |
| 2021 | 348.81 | 112°50′18″ | 31°32′37″ | 9.29 | 11.94 | 77.80 | 76.18 |
| 2022 | 347.24 | 112°42′22″ | 31°16′30″ | 9.28 | 11.92 | 76.57 | 76.14 |
| 2023 | 356.18 | 113°0′13″ | 31°37′59″ | 9.42 | 12.03 | 78.30 | 77.07 |

In the same year, the Wenchuan earthquake triggered reconstruction efforts, which also boosted transportation volumes [44], particularly after the development of the Chengguan high-speed railway. These events significantly influenced the aggregation of passenger throughput, inducing an eastward shift in the ellipse.

The direction of passenger throughput agglomeration remained stable, with the primary distribution direction expanding. Over the five-year period, the aggregation axis consistently pointed from the northeast to the southwest. From 2005 to 2009, the rotation angle of the axis decreased by 9.92°, and the ratio of the two axes of the ellipse decreased by 1.34, indicating that passenger throughput was expanding in the northeast-southwest direction while contracting in the northwest-southeast direction.

### 3.2. Characterization of the evolution of the national passenger throughput pattern, 2010–2014

From 2010 to 2014, China's economy transitioned into a post-crisis economic rebalancing phase, characterized by a shift in economic structure and a moderated GDP expansion. Despite the deceleration in the growth rate of international trade, China remained a global manufacturing and trade hub, ensuring that the air passenger market still sustained robust throughput volumes.

Between 2010 and 2014, the center of gravity of national passenger throughput shifted from 114°12′18″E, 31°43′57″N to 113°36′16″E, 31°40′48″N. As shown in Fig 3, the center of gravity initially moved northwestward from 2010 to 2012. However, in 2013–2014, the trajectory diverged southwestward, and the center of gravity moved southwestward.

In 2013, the center of gravity of passenger throughput began to shift southwest. This shift can primarily be attributed to the natural disasters in the southwest region, which caused significant damage to transportation infrastructure. In response, the government increased investment and support for infrastructure development in this area, improving transportation accessibility. These improvements contributed to the movement of civil aviation passenger throughput toward the southwest.

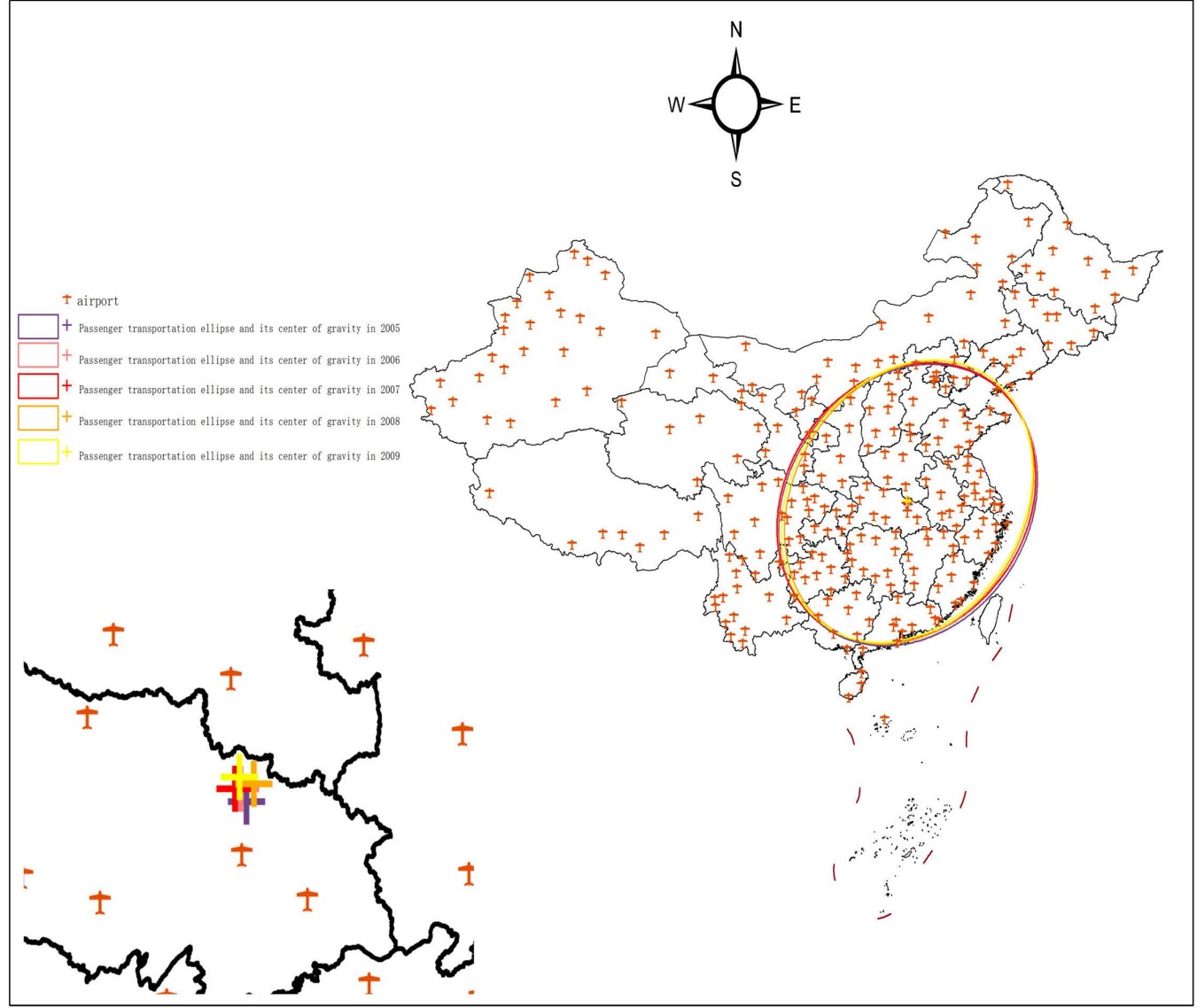

**Fig 2. National passenger throughput standard deviation ellipse and its center of gravity, 2005-2009.** The basic map came from the official website of Standard map of China at http://bzdtchinnr.gov.cn/. The drawing approval number is GS (2022) 4313. The data used was calculated by the author.

At the same time, the Chengdu-Melbourne route, the first international Oceania route in the southwest, was launched, along with the Rong-Europe International Fast Rail Freight, the fastest domestic rail freight train to Europe. The opening of the Rong-Europe Express Railway marked a significant turning point, ending the historical reliance of inland cities on ports for outward-oriented economic development. This railway helped establish Chengdu as a strategic logistics nexus in the international logistics corridor, extending to Europe, Central Asia, Russia, and ASEAN countries, supported by a stable railway channel linking China and Europe [45].

From 2010 to 2014, the scope of passenger throughput concentration showed a trend toward centralization. As shown in Table 1, the area of the standard deviation ellipse of passenger throughput increased from 310.90 km² in 2010 to

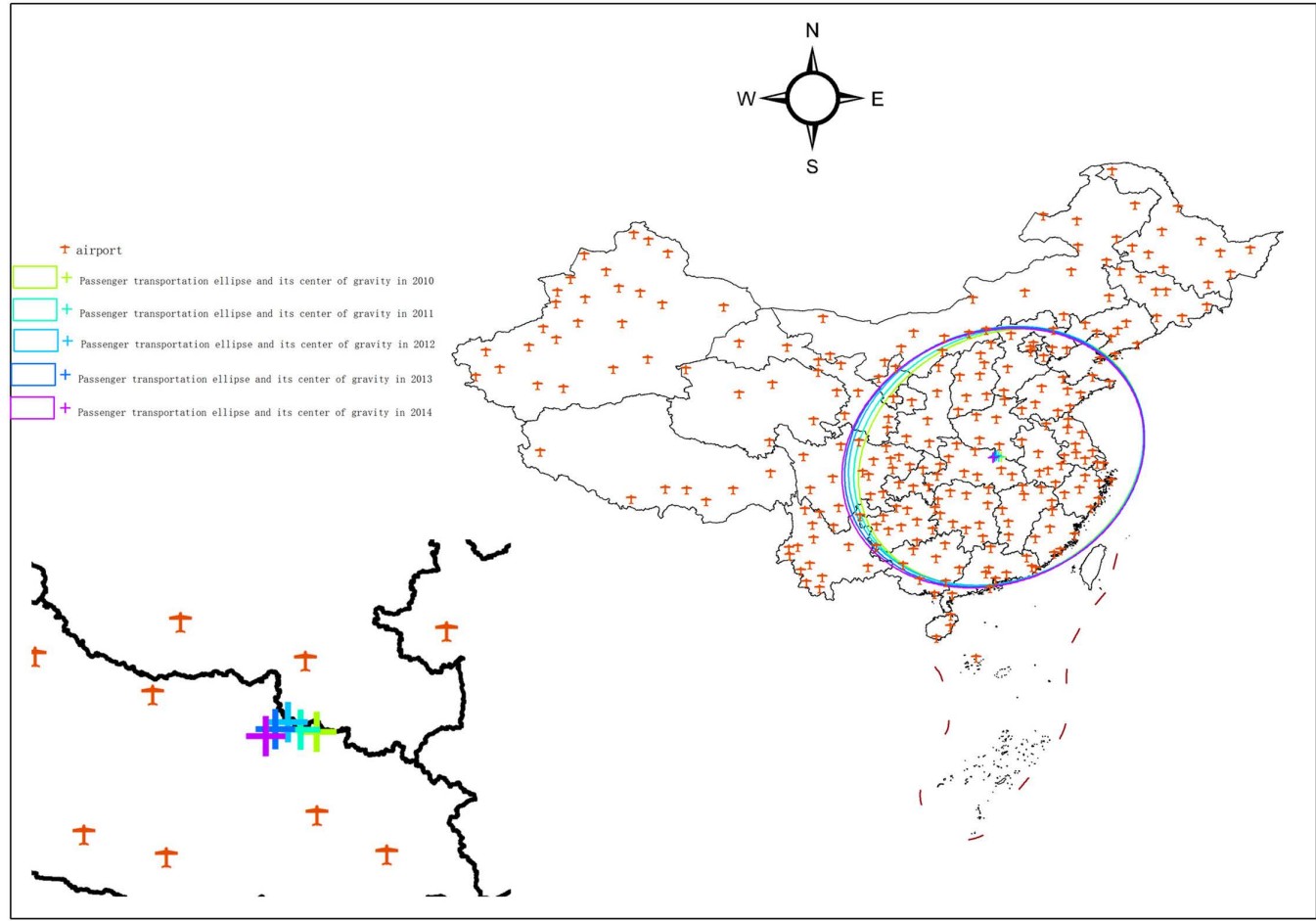

**Fig 3. National passenger throughput standard deviation ellipse and its center of gravity, 2010-2014.** The basic map came from the official website of Standard map of China at http://bzdtchinnr.gov.cn/. The drawing approval number is GS (2022) 4313. The data used was calculated by the author.

333.99 km² in 2014, expanding by 23.9 km². This indicates that passenger throughput distribution became more centralized, with an expansion towards the southwest.

The direction of passenger throughput concentration remains stable, and the main distribution direction continues to tilt towards the northeast-southwest axis. From 2010 to 2014, the rotation angle of the standard deviation ellipse increased by 7.52°, and the ratio of the two axes increased by 0.77, indicating an expansion in the northwest-southeast direction. From 2011 to 2014, the ratio of the two axes decreased by 1.44, indicating that the distribution of airports further expanded in the northeast-southwest direction.

### 3.3. Characterization of the evolution of the national passenger throughput pattern, 2015–2019

In 2015, China institutionalized the "Belt and Road Initiative (BRI)" initiative, aimed at fostering closer ties between China and regions such as Southwest Asia, Central Asia, Southeast Asia, and others. This policy promoted increased regional exchanges, business activities, and tourism, while enhancing economic cooperation with neighboring countries. A key component of this initiative was the development of the western regions of China, including improvements in transportation infrastructure such as airports, highways, and railways.

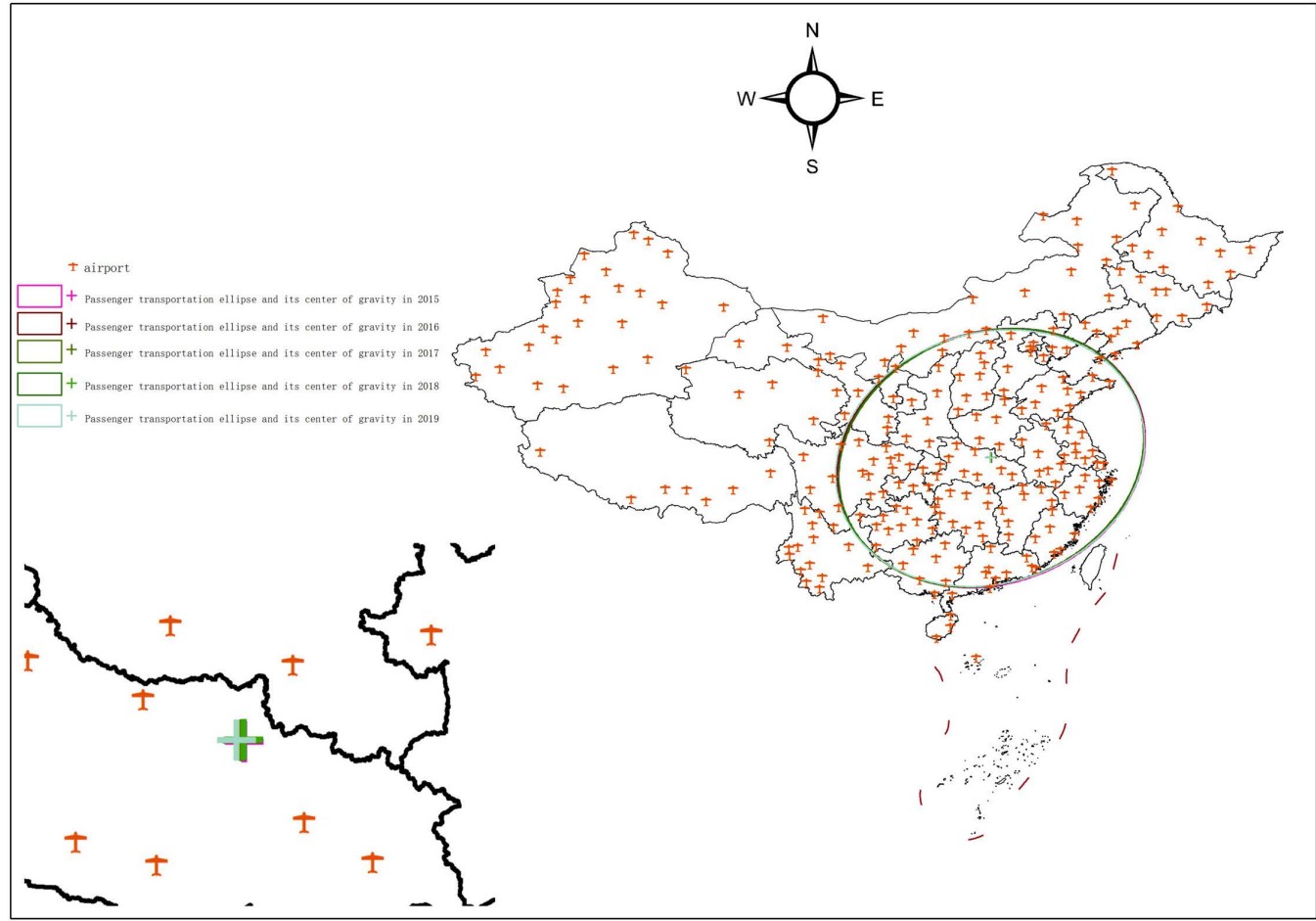

**Fig 4. National passenger throughput standard deviation ellipse and its center of gravity, 2015-2019.** The basic map came from the official website of Standard map of China at http://bzdtchinnr.gov.cn/. The drawing approval number is GS (2022) 4313. The data used was calculated by the author.

By 2017, the standard deviation ellipse revealed a clear spatial agglomeration phenomenon, signifying structural realignment in passenger flow spatial allocation. This period saw rapid development of China's high-speed rail network, which greatly improved transportation accessibility. The development of transportation infrastructure in the western regions—especially in airports, air routes, highways, and railroads—helped to reduced intercity travel duration between cities, making travel more convenient. As a result, there was an increasing competition between high-speed rail and aviation in the middle-distance market, with high-speed rail having a modal dominance in the short-distance market, while aviation maintained dominance in the long-distance market [46].

As shown in Table 1 and Fig 4, the passenger transportation center moved from 113°29′22"E, 31°41′3"N in 2015 to 113°23′9"E, 31°40′20"N by 2019, reflecting the changes in air passenger distribution and the effects of the infrastructure improvements during this period.

The range of passenger throughput agglomeration exhibited a pattern of initial concentration followed by dispersion. In 2015, the standard deviation ellipse area of passenger throughput was 340.43 km². By 2017, this area had decreased to 335.89 km², a reduction of 4.54 km², indicating that the distribution of passenger throughput became more concentrated. However, by 2019, the area increased to 338.79 km², a rise of 2.9 km² compared to 2017, signaling a shift towards more dispersed distribution.

The direction of passenger throughput agglomeration remained stable, with a tendency for the distribution to concentrate in the main direction. The rotation angle increased by 2.19° from 2015 to 2019, and the ratio of the two axes of the standard deviation ellipse decreased by 0.5 over the same period. This suggests that the distribution of passenger throughput expanded in the northeast-southwest direction, while it contracted in the northwest-southeast direction.

### 3.4. Characterization of the evolution of the national passenger throughput pattern in 2020–2023

In 2020, the outbreak of the novel coronavirus in China and its subsequent spread accelerated, leading to a clustering effect in the country's passenger traffic. However, air passenger traffic plummeted sharply due to the epidemic. In response, the government implemented stringent measures to control the spread of the virus.

By 2021, while the domestic pandemic situation ameliorated, the global situation remained serious. As a result, pandemic prevention and control within China became normalized, and air passenger traffic had not yet fully recovered [47]. In 2022, the clustering effect persisted. This year also saw the emergence of more virulent virus strains, which spread rapidly. These developments prompted even stricter epidemic control measures, leading to further declines in national passenger traffic [48].

The national drop in passenger throughput was primarily due to domestic measures such as travel restrictions and flight cancellations. Major airports in eastern China, such as Beijing Capital International Airport [49] and Shanghai Pudong International Airport, were significantly impacted. These airports, which handle a large number of international and connecting flights, saw a sharp decline in traffic due to the reduction in flight routes and the suspension of international flights, especially cross-border travel.

As shown in Table 1 and Fig 5, the center of gravity of national passenger traffic shifted from 112°59′51″E, 31°15′38″N in 2020 to 113°0′13″E, 31°37′59″N in 2023.

The center of gravity for passenger traffic gradually shifted toward the southwest, largely due to the objectives outlined in China's 14th Five-Year Plan (2021–2025), which emphasized the economic development and infrastructure improvement in the southwestern region [50]. As part of this plan, the government increased investments in transportation and tourism, significantly boosting the region's overall development. Enhanced infrastructure and the region's rich natural and cultural landscapes attracted both domestic and international tourists, leading to a rise in air passenger demand. Consequently, passenger traffic began shifting toward the southwest.

However, in 2023, following the official end of the COVID-19 pandemic, the center of gravity shifted again toward the northeast. This was likely influenced by capacity rationalization strategies taken by airlines to control the number of flights between regions affected by the virus. These restrictions were implemented to ensure necessary travel while mitigating the spread of the epidemic. As the situation improved and the virus was brought under control, it was anticipated that these measures would be relaxed, allowing for greater flexibility in flight schedules [51].Overall, the spatial agglomeration of passenger throughput followed a pattern of initial concentration followed by dispersion. The direction of this agglomeration remained stable, with a tendency for expansion in the primary direction.

To summarize, an analysis of China's national air passenger volume from 2005 to 2023 reveals a significant decline in passenger numbers in 2020, followed by a notable rebound in 2021. The year 2020 exhibited clustering effects despite the reduction in passenger volume, deviating from the usual growth patterns. This anomaly is attributed to the occurrence of unexpected public events and the implementation of related policies. Therefore, examining the impact of these government-issued policies during such events on air passenger transportation is crucial for informing the formulation and execution of future policies.

## 4. Analysis of policy effects

Analyzing the epidemiological inflection points of the COVID-19 pandemic in 2020–2021 offers valuable insights into the universal effects of China's public health policies on air transportation. Following the outbreak, the Chinese government

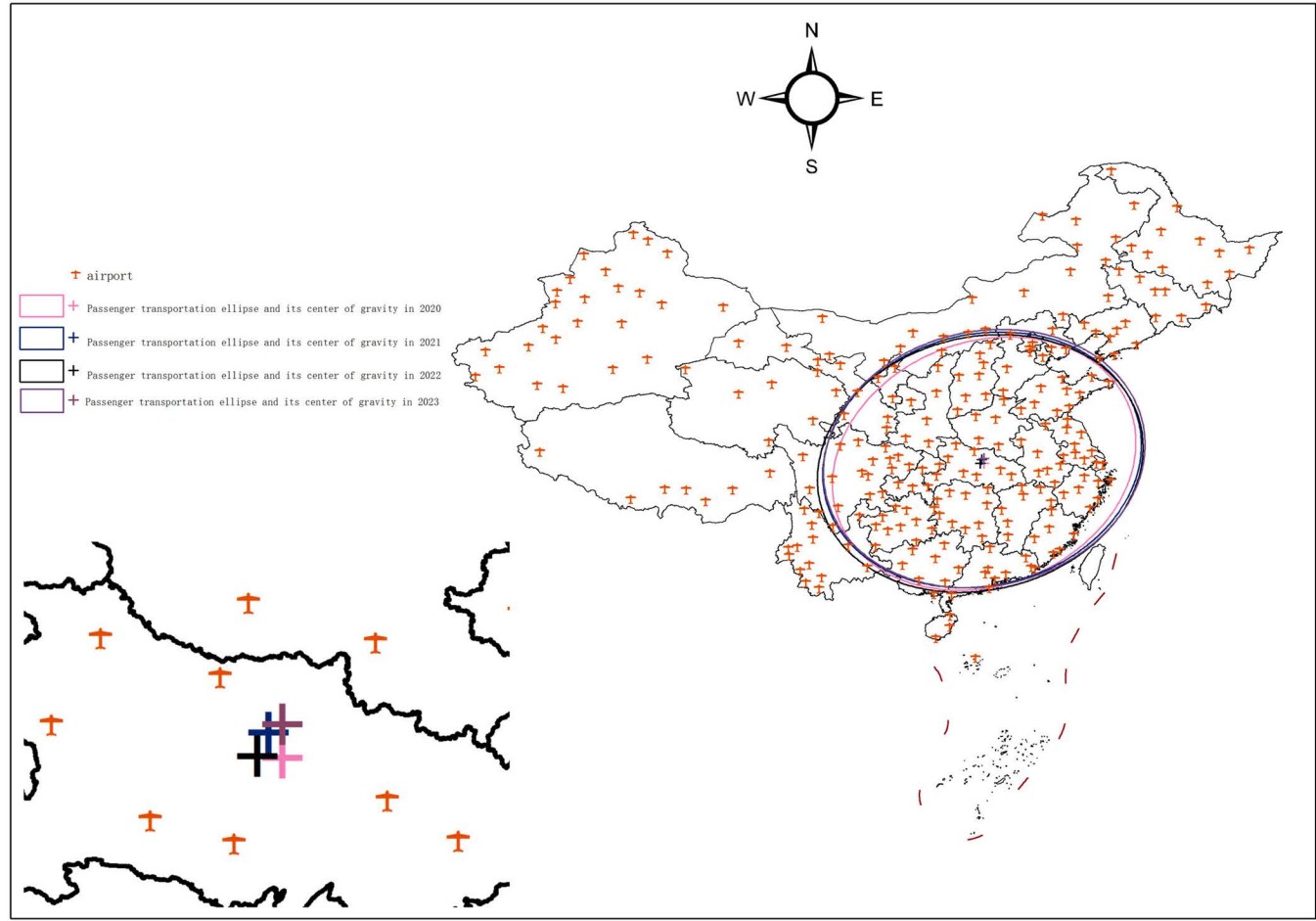

**Fig 5. National passenger throughput standard deviation ellipse and its center of gravity, 2020-2023.** The basic map came from the official website of Standard map of China at http://bzdtchinnr.gov.cn/. The drawing approval number is GS (2022) 4313. The data used was calculated by the author.

swiftly implemented a series of pandemic control measures that significantly affected air travel and passenger mobility during different periods [52].

In January 2020, Wuhan's lockdown and the nationwide implementation of stringent containment measures led to an almost complete halt in domestic air transportation. During the initial phase, strict traffic controls caused passenger volumes to plummet. Hubei Province's air passenger flow nearly ceased, while other provinces experienced significant declines.

As the pandemic came under control, policies gradually shifted from "lockdown" to "orderly resumption of work and production" and "normalized pandemic prevention." Beginning in April 2020, the State Council issued guidelines clarifying responsibilities for pandemic prevention and steadily restoring socio-economic activities, creating favorable conditions for the recovery of the air transportation sector [53].

Recovery rates varied across regions. Eastern coastal areas, where work resumed earlier, saw faster recovery, whereas central and western regions faced slower rebounds due to ongoing restrictions on mobility. Local policy innovations also exerted heterogeneous impacts in regional recovery. For instance, Guangdong employed digital tracking systems, while Guizhou implemented rapid testing measures, empirically validating the spatio-temporal evolution of regional air transportation during the pandemic.

This paper further employs a DID model to analyze the heterogeneity of pandemic control policies' impacts on air passenger volumes across time and space. This approach quantifies the effects of normalized pandemic control policies on urban economic activity, public health, and social stability, providing a foundation for evaluating policy effectiveness.

## 4.1. Benchmark regression analysis

In line with existing work [54–57], we include a set of control variables to capture their potential role in air passenger transportation and policy implications. Specifically, GDP reflects the overall level of regional economic activity, which has important implications for air passenger transportation demand. In addition, we included population size to measure the impact of population distribution on air travel. Meanwhile, airport movements were used as a proxy variable for transportation supply to control for changes in air service capacity. Finally, we also consider the potential impact of tourism revenues, tourist arrivals, and the number of hotels on air passenger movements, as these factors may significantly shape regional air transportation demand and its spatial distribution.

Regression analysis, a statistical technique, was employed to examine and quantify the relationship between dependent and independent variables. It enables us to test whether a specific independent variable significantly affects the dependent variable and to identify which variables exert the greatest influence. By analyzing regression coefficients, we can better understand the patterns and trends within the data, as the coefficients directly reflect the impact of each independent variable on the dependent variable.

The results from the benchmark regression, presented in Table 2 (columns 1–4), demonstrate that all coefficients are positive, significant, and pass the 1% significance level, whether or not control variables are included and individual and time effects are accounted for. Critically, the coefficients in columns (2), (3), and (4) differ significantly from those in column (1). The difference arises because column (1) does not include control variables. For air passenger traffic, the impact of related policies is influenced by the geographic location and administrative level of the host city. Furthermore, column (4) controls for both individual and time fixed effects. After including all control variables, the robust standard error decreases and the coefficient increases by 0.815, passing the 1% significance test. This indicates that the results obtained from using a double fixed effects model (individual and time) are more robust and reliable.

## 4.2. Parallel trend tests

To ensure the accuracy and reliability of the benchmark regression results, it is essential to test for parallel trends. The validity of the benchmark model heavily depends on whether the assumption of parallel trends holds true. Fig 6 presents the results of the parallel trend test, which evaluates whether the trends in the data before the policy intervention were consistent across heterogeneous groups. This test is crucial for confirming that any observed effects can be attributed to the policy intervention rather than pre-existing differences in trends between groups.

The confidence interval includes zero prior to the implementation of the policy, indicating no significant difference in trends between the groups. However, after the second year of policy implementation, the confidence interval no longer includes zero, showing a significant deviation from the pre-policy trend. This confirms that the parallel trend assumption holds.

The policy intervention exhibited a temporal lag effect. In the first year after its implementation, there was no significant recovery in passenger traffic in the affected cities. This delay can be attributed to institutional inertia during bureaucratic adaptation periods, policy enforcement, and the reallocation of resources. By the second year, the policy began to show a tangible effect on passenger traffic, with the impact peaking in that period.

## 4.3. Robustness tests

**4.3.1. Placebo test.** Robustness diagnostics constitute a critical step in empirical analysis to validate the reliability of the empirical analysis conclusions. To mitigate the potential influence of unobservable, uncontrollable, and time-varying individual characteristics on the estimation results, a placebo test is conducted. This entails implementing a city-year

stratified randomization procedure using randomly selected cities and years, followed by performing a benchmark regression. The process is repeated 500 times to generate a distribution of the estimated coefficients for T. The results are evaluated by comparing the probability of obtaining the baseline regression estimates from the randomized experiment, enabling the detection of unobserved heterogeneity or stochastic noise such as omitted variables or random factors.

The results in Fig 7 indicate that the P coefficients are centered around 0 under the randomized treatment, confirming that the benchmark model is free from specification errors like omitted variables, and satisfies the diagnostic criteria the placebo test.

**4.3.2. Shrink-tailed regression.** To mitigate the impact of outliers in the dependent variable in the explanatory variable, passenger traffic, on the regression results, a winsorized regression was employed. The results presented in Table 3 demonstrate that, even with tail shrinking at the 1%-99% and 5%-95% levels, the number of landings and takeoffs significantly affect passenger traffic volume. Furthermore, the policy impact coefficients show only a minimal deviation when compared to the baseline regression, indicating that the model passes the shrink-tailed test.

**4.3.3. Narrowing the sample.** To mitigate the impact of the 2008 international financial crisis and China's macro-policy of implementing the 4 trillion yuan plan, the sample time range for the benchmark regression was limited to 2010–2022. As shown in column (2) of Table 3, after narrowing the sample period, the sign and significance of the T-estimate coefficients remain unchanged. Additionally, the number of landings and takeoffs continues to have a significant effect on passenger traffic in the city, confirming the robustness of the results.

**Table 2. Benchmark regression results on the impact of public health emergency prevention and control policies on air passenger transportation.**

|  | (1) | (2) | (3) | (4) |
|---|---|---|---|---|
| T | 1.087*** | 0.796*** | 0.703*** | 0.815*** |
|  | (8.91) | (5.85) | (6.96) | (6.02) |
| G |  | −0.025 | −0.019 | −0.011 |
|  |  | (−0.85) | (−0.69) | (−0.39) |
| RP |  | 0.017 | −0.033** | 0.009 |
|  |  | (1.41) | (−2.16) | (0.63) |
| CP |  | −0.054 | 0.025 | −0.019 |
|  |  | (−1.49) | (0.63) | (−0.54) |
| CM |  | 0.154*** | 0.139*** | 0.167*** |
|  |  | (3.05) | (2.78) | (3.42) |
| FP |  | 0.026 | −0.053 | 0.021 |
|  |  | (0.43) | (−0.99) | (0.37) |
| FM |  | 0.172** | 0.224** | 0.162** |
|  |  | (2.31) | (2.41) | (2.33) |
| H |  | −0.015 | 0.020 | −0.015 |
|  |  | (−1.08) | (1.07) | (−1.01) |
| constant | −0.000** | −0.000*** | −0.000 | −0.000*** |
|  | (−2.48) | (−2.73) | (−0.00) | (−3.23) |
| Time Fixed | Yes | No | Yes | Yes |
| Individual Fixed | Yes | Yes | No | Yes |
| N | 2512 | 2512 | 2512 | 2512 |
| R2 | 0.974 | 0.981 | 0.941 | 0.982 |

Note: () are robust standard errors, ***, ** represent significance levels of 1% significant, 5% significant, in that order, and the same table below.

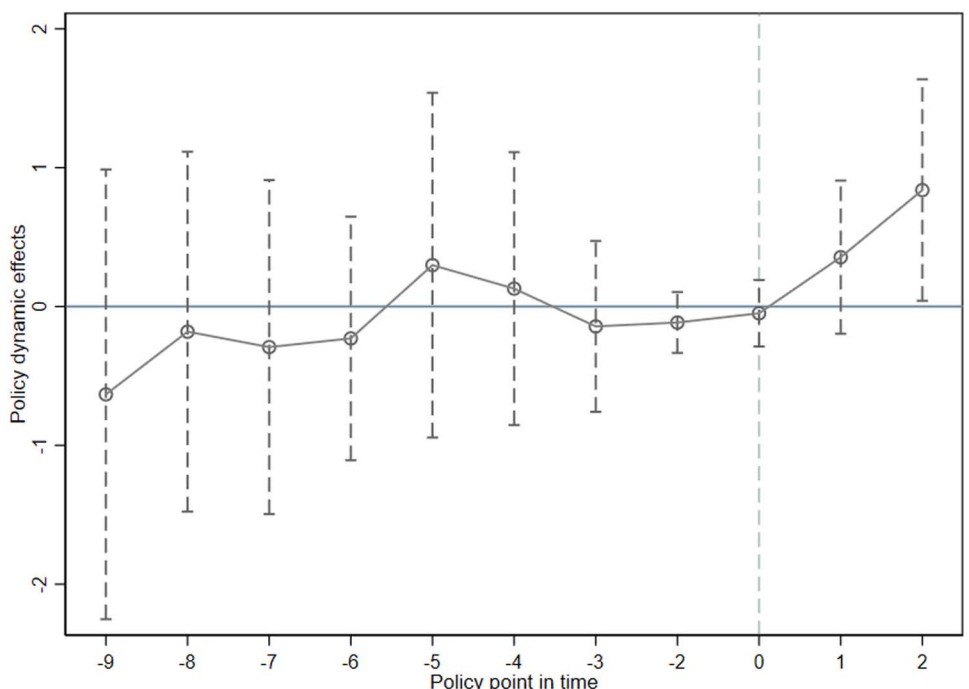

**Fig 6. Parallel trend test results of the impact of public health emergency prevention and control policies on air passenger transportation.**

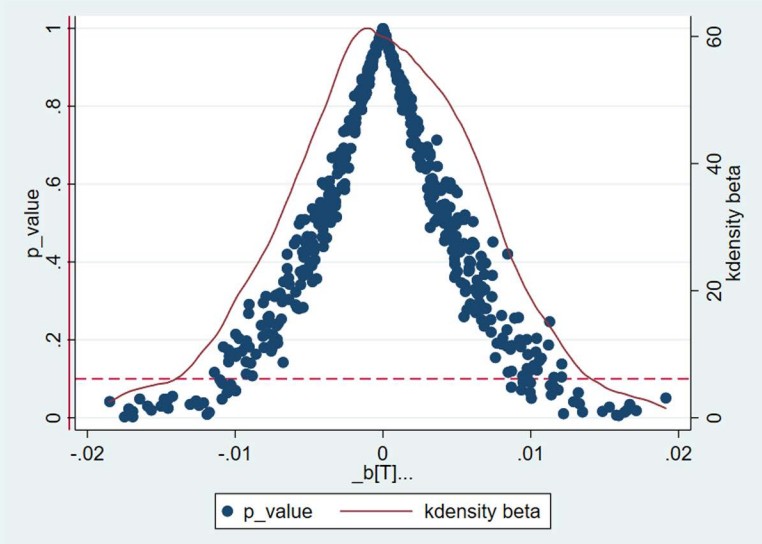

**Fig 7. Placebo test results for the impact of public health emergency prevention and control policies on air passenger transportation.**

**4.3.4. Lag test.** The lag test is a statistical technique used to detect whether there is a delay in the effect of one variable on itself or on other variables in time series data. It helps identify and quantify such delayed effects, allowing for the construction of more accurate predictive models and a better understanding of causal relationships and dynamics between variables.

**Table 3. Robustness test results of the impact of public health emergency prevention and control policies on air passenger transportation.**

| | (1)<br>Downsizing 1%-99% | (2)<br>Downsizing 5%-95% | (3)<br>Narrowing the sample interval | (4)<br>Control variables lagged one periods | (5)<br>Control variables lagged two periods | (6)<br>Control variables lagged three periods |
|---|---|---|---|---|---|---|
| T | 0.631*** | 0.394*** | 0.944*** | 0.949*** | 1.024*** | 1.133*** |
| | (4.55) | (3.03) | (8.9861) | (6.26) | (6.50) | (8.56) |
| Control variables | Yes | Yes | Yes | Yes | Yes | Yes |
| Individual Fixed | Yes | Yes | Yes | Yes | Yes | Yes |
| Time Fixed | Yes | Yes | Yes | Yes | Yes | Yes |
| constant | −0.015 | −0.085*** | −0.022*** | −0.006 | −0.014* | −0.029*** |
| | (−1.23) | (−3.76) | (−4.3016) | (−1.41) | (−1.91) | (−2.94) |
| N | 2272 | 1388 | 1810 | 2321 | 2150 | 1988 |
| $R^2$ | 0.963 | 0.933 | 0.988 | 0.980 | 0.981 | 0.982 |

Note: () are robust standard errors, ***, * represent significance levels of 1% significant, 10% significant, in that order, and the same table below.

In this analysis, the lags of the control variables were adjusted, and one, two, and three-period lags were incorporated into the double difference model for the benchmark regression. As shown in columns (4), (5), and (6) of Table 3, the T coefficients maintain the same sign and significance as in the baseline regression. These results, along with the previous tests, empirically validate the robustness of the conclusions.

### 4.4. Heterogeneity analysis

**4.4.1. Geographic location heterogeneity analysis.** To assess the impact of geographic location on the policy effects of public health events, a heterogeneity test is conducted by dividing the country into four regions: East, Central, West, and Northeast. Table 4, columns (1) to (4), show the local T-coefficients for each region, listed in order of their magnitude: East, Central, West, and Northeast.

Firstly, the eastern coastal region is economically developed, with well-established infrastructure and relatively sufficient public health resources. As a result, the region exhibited a more rapid policy response to the public health policies, leading to a more pronounced recovery in passenger traffic. Moreover, prior to the implementation of the policy, the eastern region already had some of the highest passenger traffic in the country, and the policy exacerbated pre-existing regional disparities;Secondly, the central region also benefited from policy support, but the response speed and intensity were not as fast as in the eastern region. Consequently, the increase in passenger traffic was smaller compared to the eastern region;Thirdly, the western region, being economically less developed and having fewer public health resources, experienced a more pronounced marginal effect of the policy. This is because the region had higher marginal returns to policy intervention, making the policy impact more noticeable;Finally, the northeastern region is undergoing an economic transition, and its public health resources and response capabilities are limited. Therefore, the implementation and impact of the policy were slower, and the full potential of the policy was not realized due to the region's relatively lagging market economy.

In summary, these regional differences in policy effects underscore how varying levels of economic development, infrastructure, and public health resources influence the outcomes of public health policies.

**4.4.2. Heterogeneity analysis of city administrative hierarchy.** In this study, cities are divided into two categories: provincial capitals and higher-tier cities, and other cities. Two benchmark regressions are then conducted. The results presented in columns (5) and (6) of Table 4 indicate that the implementation of public health emergency policies has a significant positive effect on the passenger traffic volume of host cities. However, the policy's impact is more pronounced

**Table 4. Heterogeneity results of geographic location and city administrative level.**

|  | (1) | (2) | (3) | (4) | (5) | (6) |
|---|---|---|---|---|---|---|
|  | **Eastern part** | **Central section** | **Western part** | **North-east** | **Provincial capitals and higher-tier cities** | **Otherwise** |
| T | 1.152*** | 0.593*** | 0.494** | 0.467* | 1.223*** | 0.251*** |
|  | (13.24) | (3.36) | (2.49) | (1.82) | (25.92) | (2.74) |
| Control variables | Yes | Yes | Yes | Yes | Yes | Yes |
| Individual Fixed | Yes | Yes | Yes | Yes | Yes | Yes |
| Time Fixed | Yes | Yes | Yes | Yes | Yes | Yes |
| constant | −0.073*** | −0.146*** | 0.123*** | −0.023 | −0.313*** | −0.158*** |
|  | (−2.91) | (−4.02) | (3.73) | (−0.21) | (−4.83) | (−6.29) |
| N | 797 | 489 | 946 | 280 | 538 | 1973 |
| $R^2$ | 0.993 | 0.956 | 0.972 | 0.967 | 0.995 | 0.954 |

Note: () are robust standard errors, ***, **, and * represent significance levels of 1% significant, 5% significant, and 10% significant, in that order, and the same table below.

in provincial capitals and cities at or above the provincial capital level, where passenger traffic recovery is more substantial.

This can be attributed to the fact that provincial capitals and cities at this level are typically the political, economic, and cultural centers of their respective provinces. These cities are equipped with abundant public health resources and have stronger policy implementation capabilities. In the face of public health emergencies, provincial capitals can implement policies more swiftly due to their high administrative capacity and greater ability to mobilize resources. Additionally, their infrastructure and public services are more comprehensive and robust compared to other cities, making them better positioned to maintain the development of air passenger transportation at the forefront.In contrast, while other cities also benefit from policy interventions, they tend to have fewer resources and less robust policy implementation. As a result, these cities face greater challenges during public health emergencies, and the implementation of policies is often less effective than in provincial capitals. Consequently, the impact on the recovery of passenger transportation is relatively smaller, and these cities encounter challenges in attaining the same level of recovery as the provincial capitals.

In summary, significant differences in policy effectiveness emerge between regions and cities when responding to public health emergencies. These differences stem from variations in economic development, infrastructure, and public health resources. Eastern coastal regions and provincial capitals exhibit stronger policy responses and faster recovery of passenger traffic. To improve the overall effectiveness of policies, it is recommended that regional disparities be carefully considered during policy formulation, with additional support provided to central and western regions as well as non-provincial capital cities. Furthermore, enhancing regional coordination and promoting resource sharing can help reduce imbalances in policy outcomes.

## 5. Conclusion

This study employs the SDE method combined with ArcGIS visualization to analyze the spatiotemporal evolution of air passenger traffic in China from 2005 to 2023. The results indicate that the overall distribution of air passenger traffic in China exhibits a northeast-southwest spatial axis, with an increasing trend over time. The center of the standard deviational ellipse gradually shifted from the northwest to the southwest and then moved back northeast in 2023. Furthermore, changes in the area of the standard deviational ellipse in 2008, 2017, 2020, and 2022 demonstrate spatial polarization phenomena in air passenger traffic. Notably, air passenger traffic and clustering intensity increased significantly in 2008 and 2017, whereas a sharp decline was observed in 2020 and 2022 due to public health emergencies. This trend

indicates a positive correlation between air passenger throughput and public health emergencies, while also highlighting the effectiveness of public health emergency policies in regulating air passenger mobility [58,59].

Additionally, this study employs a DID model to assess the impact of passenger flow guidance policies on air passenger traffic. The findings reveal that the degree of recovery varied across cities due to differences in population size, resource allocation, and healthcare infrastructure. Significant regional disparities exist in responses to national policies among eastern, central, western, and northeastern China. Furthermore, provincial capitals and cities at or above the prefecture level exhibited a faster recovery in air passenger traffic compared to other cities, emphasizing the role of regional differences in policy effectiveness. The results suggest that passenger flow guidance policies had the most pronounced impact in the eastern region, whereas the recovery of air passenger traffic in the central, western, and northeastern regions was relatively slower.

These regional disparities underscore the necessity of future policy adjustments tailored to regional characteristics. For instance, in the eastern region, where air travel demand is high, increasing route density and flight frequency could better meet market needs. Meanwhile, in the central, western, and northeastern regions, strengthening infrastructure investment and providing policy subsidies could attract more air traffic and foster their development as transportation hubs. Additionally, this study recommends establishing a regular policy evaluation mechanism to dynamically adjust aviation policies based on fluctuations in air passenger traffic. However, this study also acknowledges its limitations, as it does not discuss specific flight routes or schedules. Future research could further explore these aspects to provide more comprehensive insights into regional civil aviation development.

## Author contributions

**Data curation:** Chen Luo, Min Wang, Xinrong Hu, Chunyang Liu, Xutong Wang.

**Formal analysis:** Chen Luo, Tianshun Ma, Min Wang, Xinrong Hu, Chunyang Liu, Xutong Wang.

**Investigation:** Chen Luo.

**Resources:** Tianshun Ma, Min Wang, Xinrong Hu, Chunyang Liu, Xutong Wang.

**Software:** Tianshun Ma.

**Writing – original draft:** Chen Luo.

**Writing – review & editing:** Chen Luo, Tianshun Ma.

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
