## [Decision Letter · Decision Letter 0]

14 Oct 2024

PONE-D-24-36622Analysis of the temporal and spatial evolution of china's air passenger transportation and the impact of policies on passenger flowsPLOS ONE

Dear Dr. LUO,

Thank you for submitting your manuscript to PLOS ONE. After careful consideration, we feel that it has merit but does not fully meet PLOS ONE’s publication criteria as it currently stands. Therefore, we invite you to submit a revised version of the manuscript that addresses the points raised during the review process.

We look forward to receiving your revised manuscript.

Kind regards,

Qing-Chang Lu

Academic Editor

PLOS ONE

Journal Requirements:

3. We note that Figures 1-5 in your submission contain map/satellite images which may be copyrighted. All PLOS content is published under the Creative Commons Attribution License (CC BY 4.0), which means that the manuscript, images, and Supporting Information files will be freely available online, and any third party is permitted to access, download, copy, distribute, and use these materials in any way, even commercially, with proper attribution. For these reasons, we cannot publish previously copyrighted maps or satellite images created using proprietary data, such as Google software (Google Maps, Street View, and Earth). For more information, see our copyright guidelines: http://journals.plos.org/plosone/s/licenses-and-copyright. We require you to either (a) present written permission from the copyright holder to publish these figures specifically under the CC BY 4.0 license, or (b) remove the figures from your submission:

a. You may seek permission from the original copyright holder of Figures 1-5 to publish the content specifically under the CC BY 4.0 license. We recommend that you contact the original copyright holder with the Content Permission Form (http://journals.plos.org/plosone/s/file?id=7c09/content-permission-form.pdf) and the following text: “I request permission for the open-access journal PLOS ONE to publish XXX under the Creative Commons Attribution License (CCAL) CC BY 4.0 (http://creativecommons.org/licenses/by/4.0/). Please be aware that this license allows unrestricted use and distribution, even commercially, by third parties. Please reply and provide explicit written permission to publish XXX under a CC BY license and complete the attached form.” Please upload the completed Content Permission Form or other proof of granted permissions as an "Other" file with your submission. In the figure caption of the copyrighted figure, please include the following text: “Reprinted from [ref] under a CC BY license, with permission from [name of publisher], original copyright [original copyright year].”

b. If you are unable to obtain permission from the original copyright holder to publish these figures under the CC BY 4.0 license or if the copyright holder’s requirements are incompatible with the CC BY 4.0 license, please either i) remove the figure or ii) supply a replacement figure that complies with the CC BY 4.0 license. Please check copyright information on all replacement figures and update the figure caption with source information. If applicable, please specify in the figure caption text when a figure is similar but not identical to the original image and is therefore for illustrative purposes only. The following resources for replacing copyrighted map figures may be helpful: USGS National Map Viewer (public domain): http://viewer.nationalmap.gov/viewer/ The Gateway to Astronaut Photography of Earth (public domain): http://eol.jsc.nasa.gov/sseop/clickmap/ Maps at the CIA (public domain): https://www.cia.gov/library/publications/the-world-factbook/index.html and https://www.cia.gov/library/publications/cia-maps-publications/index.html NASA Earth Observatory (public domain): http://earthobservatory.nasa.gov/ Landsat: http://landsat.visibleearth.nasa.gov/ USGS EROS (Earth Resources Observatory and Science (EROS) Center) (public domain): http://eros.usgs.gov/# Natural Earth (public domain): http://www.naturalearthdata.com/

Additional Editor Comments:

There are important comments and suggestions from the reviewers. The manuscript should be carefully revised significantly.

Reviewers' comments:

Reviewer's Responses to Questions

**Comments to the Author**

1. Is the manuscript technically sound, and do the data support the conclusions?

Reviewer #1: No

Reviewer #2: Yes

2. Has the statistical analysis been performed appropriately and rigorously? 

Reviewer #1: No

Reviewer #2: I Don't Know

3. Have the authors made all data underlying the findings in their manuscript fully available?

Reviewer #1: Yes

Reviewer #2: Yes

4. Is the manuscript presented in an intelligible fashion and written in standard English?

Reviewer #1: No

Reviewer #2: No

5. Review Comments to the Author

Reviewer #1: 1. The abstract fails to clearly express the importance and value of the research

2. The logic and coherence of sentences need to be improved, and professional academic terms need to be clear

3. Only two important conclusions through the whole study, which is weak and cannot support practical reference significance

4. The introduction needs a big improvement. After reading this chapter, it is not clear to me why the author studies this topic, why through the methods used by the author, why these are necessary, and why these things can be published in international journals

5. Literature review also needs to be carefully revised. The existing summary and induction are very immature, and the academic expression also needs to be improved. In this part, the author should focus on showing sufficient insight on the research subject area.

6. Why are there Chinese characters in the picture?

7.This research needs to do more and lacks explanation and basis for the application of research methods, also need to add the discussion chapters in order to present their academic value from comparation to the related state of art works.

Reviewer #2: This paper analyzes the evolution characteristics of China's air passenger traffic from 2005 to 2023 and presents a substantial amount of work.

Review comments are as follows:

1. Please carefully check the formatting and language issues in the article. For example, the word "China's" in the title should have its first letter capitalized; the first letter of the line below Fig. 4 should also be capitalized; there are grammatical issues in the first sentence of section 3.1.

2. The images are unclear, and legends need to be presented in English. Please make the necessary revisions throughout the text.

3. How does the second paragraph of the introduction relate to the article's title? Please adhere to scientific writing standards and avoid unrelated descriptions.

4. The last paragraph of the introduction mentions, “Meanwhile, the double difference model, as a method of assessing the impact of policies and other factors on a certain variable, has been widely used in the field of high-speed rail, but less so in air transportation.” Is the Double Difference Model applicable to the aviation sector?

5. The description in the second chapter is too vague and does not closely relate to the main theme of the article. In fact, the paper does not reflect specific differences between policies. Please seriously consider whether the Policy Background is necessary to include in the paper.

6. The methodology in section 3.3.1 lacks explanations for the parameters.

7. Why was the Double Difference Model chosen? Are there other models that could address this issue? What are the advantages of the Double Difference Model compared to others?

8. The variation of the center of gravity in the figures is not apparent; it is recommended to zoom in on the image to highlight the trend of changes in the center of gravity.

6. PLOS authors have the option to publish the peer review history of their article (what does this mean? ). If published, this will include your full peer review and any attached files.

**Do you want your identity to be public for this peer review?** For information about this choice, including consent withdrawal, please see our Privacy Policy .

Reviewer #1: No

Reviewer #2: No

---

## [Author Response · Author response to Decision Letter 1]

26 Dec 2024

Response to Reviewer #1

Response:

We thank the referee very much for the feedback and nice suggestions. In this revision, we have followed the suggestions to revise the paper accordingly, with all the revisions highlighted with red color.

1. The abstract fails to clearly express the importance and value of the research.

Response to 1

Thank you for your comments. On page 1, lines 21-24, and page 2, lines 38-42, we have summarized the importance and value of the study. This study is significant in assisting airlines to optimize their route networks and capacity allocation to adapt to dynamic passenger flow patterns, and provides important insights for policy makers to design air passenger flow guidance policies, which can help to formulate more precise and efficient measures to promote the balanced development of regional, territorial and even national passenger transportation.

2. The logic and coherence of sentences need to be improved, and professional academic terms need to be clear.

Response to 2

Thank you for pointing this out. Thorough changes have been made in the revised draft to enhance the logical flow and coherence of sentences, while ensuring the use of standardized terminology.

3. Only two important conclusions through the whole study, which is weak and cannot support practical reference significance.

Response to 3

Thank you for your valuable comments. On pages 34-35, lines 630-652 have been revised for the conclusion. First, regarding the spatial and temporal evolution of air passenger traffic, the overall pattern of China's air passenger traffic shows a northeast-southwest direction, which mainly follows this trend. However, the influence of the north-south direction gradually increases, which is manifested in the change of the ratio of the long and short axes of the standard deviation ellipse, reflecting the expansion of air passenger traffic in the north-south direction. This finding highlights the dynamic evolution of air passenger flow patterns over time, providing useful insights for airlines to optimize routes and adjust capacity. Second, the shift in the center of gravity of air passenger traffic. From 2005 to 2022, the center of gravity of air passenger traffic shows a clear trend of shifting from the northwest to the southwest, indicating that the southwest region is becoming increasingly important in China's air passenger traffic. However, in 2023, the center of gravity of passenger flow begins to move toward the northeast, signaling a new adjustment in the pattern of air passenger flow. This change is critical to airline route planning and demonstrates the adaptability of air transportation to changes in economic and regional factors. Finally, regional differences in the impact of policies on air passenger flow, there are significant differences in the impact of policies on air passenger flow in different regions. Geographically, passenger flow growth in the eastern region is significantly higher than in the central, western and northeastern regions. At the city level, provincial capitals and municipalities are more responsive to policies than other prefecture-level cities. This finding emphasizes the importance of developing regional policies based on local characteristics, such as population size, medical resources, and infrastructure development.

4. The introduction needs a big improvement. After reading this chapter, it is not clear to me why the author studies this topic, why through the methods used by the author, why these are necessary, and why these things can be published in international journals.

Response to 4

Thank you for your valuable comments. We have made detailed revisions to the introduction.

With the rapid growth of China's aviation market and the continuous adjustment of the policy environment, the strategic position of air transportation in promoting economic development and enhancing regional connectivity has become increasingly prominent. However, China's air transport system still faces many challenges, including an air network structure that needs to be optimized, unbalanced regional development, and an urgent need to improve the synergistic effectiveness with other modes of transportation such as railroads and highways. In this context, it is of great theoretical and practical value to study the spatial and temporal evolution of air passenger flows and the specific impact of passenger flow guidance policies on passenger flows. This study can not only provide a scientific basis for policy makers to guide the design of regional differentiation policies, but also provide data support for industry decision makers to optimize the layout of route networks and resource allocation. Especially for the current status quo of unbalanced regional aviation development, analyzing the heterogeneous impact of policies on regional air passenger flow can help formulate targeted measures such as supporting the construction of airports in central and western regions and improving regional subsidy policies. It also provides a valuable reference for the formulation of more flexible flight regulation programs and passenger protection measures in case of emergencies.

Difference-in-Differences (DID) model, as a classic measure of causal inference, can effectively control the interference of unobserved fixed effects and potential time trends on the dependent variable by combining the time dimension with the comparison of treatment and control groups, so as to accurately identify the causal effects of policies or events. The air transportation sector is characterized by significant spatial heterogeneity and temporal dynamics, and there are differences in the implementation effects of policies and their impacts over time in different regions, which makes the DID model an ideal tool for analyzing such issues. It has been shown that DID models have demonstrated strong applicability and robustness in transportation sectors such as railroads, highways, water transport and air transport, especially in quantifying the effects of policy interventions and revealing regional differences. Although there have been studies focusing on the development of air transportation and its policy effects, there is still a lack of systematic analysis on how policies shape the spatial and temporal distribution of air passenger flows, especially the mechanisms at play in the context of major emergencies. This study analyzes the passenger traffic data from 2005 to 2023 and finds that the passenger traffic decreases sharply in 2020 and recovers sharply in 2021, based on which we quantitatively analyze the temporal dynamics and spatial heterogeneity of the impact of the passenger flow guidance policy on air passenger flows during the period of 2020 through the introduction of the DID model.

5. Literature review also needs to be carefully revised. The existing summary and induction are very immature, and the academic expression also needs to be improved. In this part, the author should focus on showing sufficient insight on the research subject area.

Response to 5

We thank the reviewers for their valuable comments. In the literature review section we have restructured the literature review section to provide a more in-depth review of previous studies on air passenger transportation, policy implications and spatio-temporal analysis. See the Introduction section for details.

6. Why are there Chinese characters in the picture?

Response to 6

Thank you for your feedback. We apologize for this oversight. All figures have been revised in the revised version to ensure that the legends, notes and labels are in English, in line with journal standards.

7.This research needs to do more and lacks explanation and basis for the application of research methods, also need to add the discussion chapters in order to present their academic value from comparation to the related state of art works.

Response to 7

Thank you for your valuable comments. In-depth revisions have been made to address the issues you mentioned, such as the insufficient explanation and basis for the application of the research methodology.

The DID method can effectively deal with the before-and-after comparison of policy changes, control potential disturbances, and provide more accurate causal inference. In the revised draft, the theoretical basis for the selection of the research method and its application is elaborated in greater detail, the literature review section is expanded, and more research on the application of double difference models (DID) in the field of transportation is included. The Introduction and Research Methodology 2.3.2 sections are detailed, relevant descriptions are added, and research results in related fields are cited to demonstrate the effectiveness of the methodology in analyzing air passenger flows and policy impacts.

Response to Reviewer #2

 This paper analyzes the evolution characteristics of China's air passenger traffic from 2005 to 2023 and presents a substantial amount of work.

Response:

We thank the referee very much for carefully reviewing our paper and providing the very nice and constructive suggestions, which help improve the quality of this paper even further. It is our pleasure. In this revision, we have followed the suggestions to revise the paper accordingly, with all the revisions highlighted with red color.

1.Please carefully check the formatting and language issues in the article. For example, the word "China's" in the title should have its first letter capitalized; the first letter of the line below Fig. 4 should also be capitalized; there are grammatical issues in the first sentence of section 3.1.

Response to 1

Thank you for pointing out the problems. These issues have been corrected in the revised manuscript, including capitalization in titles and figure headings. The specific examples mentioned by the reviewers have been carefully addressed.

2. The images are unclear, and legends need to be presented in English. Please make the necessary revisions throughout the text.

Response to 2

Thank you for pointing this out. All charts have been updated for clarity and the legends have been revised to English.

3. How does the second paragraph of the introduction relate to the article's title? Please adhere to scientific writing standards and avoid unrelated descriptions.

Response to 3

Thank you for your suggestions. We have carefully reviewed the content of the second paragraph of the introduction and revised lines 68-95 on pages 4-5 to highlight the central role of policies in the spatial and temporal evolution of air passenger transportation, and to illustrate how these policies have shaped the spatial patterns and temporal distributions of passenger flows through the analysis of specific policies (e.g., airport construction, lowering of market access thresholds, and punctuality enhancement) so that they are more closely related to the theme of the article's title. It also removes a number of broader statements that are not sufficiently relevant to the topic of air transportation.

4. The last paragraph of the introduction mentions, “Meanwhile, the double difference model, as a method of assessing the impact of policies and other factors on a certain variable, has been widely used in the field of high-speed rail, but less so in air transportation.” Is the Double Difference Model applicable to the aviation sector?

Response to 4

I thank the reviewers for their questions. In response to the applicability of the double difference model to the aviation domain, on page 10, lines 206-214, it is demonstrated by combing through existing literature that the method can be effectively used to analyze the causal effects of external events in the aviation domain. These studies show that the double difference model has wide applicability in the aviation domain, especially in analyzing the effects of external factors on the aviation domain, and can provide clear causal inferences and quantitative results. In this study, the double difference model is also applicable in exploring the impact of the new crown epidemic prevention and control policy on air passenger traffic. On the one hand, air transportation has significant temporal and spatial heterogeneity under policy intervention; on the other hand, the panel data structure can fully satisfy the basic assumptions of the double-difference model, such as the parallel trend assumption. Through this method, we are able to accurately assess the impact of the policy and provide theoretical basis and practical support for policy optimization and industry recovery in the aviation sector.

5. The description in the second chapter is too vague and does not closely relate to the main theme of the article. In fact, the paper does not reflect specific differences between policies. Please seriously consider whether the Policy Background is necessary to include in the paper.

Response to 5

Thank you to the reviewers for their valuable feedback on the article. According to your suggestion, the policy background content of the original Chapter 2 has been revised and moved to Chapter 4, “Analysis of Policy Effects”, in order to more closely integrate with the theme of the article. Based on the analysis of air passenger traffic data from 2005 to 2023, it is found that the air passenger flow drops sharply in 2020 and shows a rebound trend in 2021, and based on the special node of 2020, the impact of the passenger flow guidance policy is quantified and analyzed. Therefore, the research of this paper takes the special year of 2020 as the entry point, and in order to reveal this impact in depth, this paper introduces the double difference model to explore the spatio-temporal heterogeneity of the impact of the passenger flow guidance policy on air passenger traffic from a quantitative point of view, which provides empirical support for evaluating the effect of the policy and optimizing the future policy.

6. The methodology in section 3.3.1 lacks explanations for the parameters.

Response to 6

Thank you for your feedback. We have added an explanation of the use of parameters on page 13, lines 267-282.

7. Why was the Double Difference Model chosen? Are there other models that could address this issue? What are the advantages of the Double Difference Model compared to others?

Response to 7

We thank the reviewers for their important questions. Difference-in-Differences (DID) modeling is a widely used analytical method in econometrics and causal inference studies, which is particularly suitable for assessing the causal effects of policy interventions or external events. Compared with other models, the DID model can effectively eliminate unobserved fixed effects and potential time trends on the dependent variable by combining the “time” dimension with the “treatment and control group” dimensions, so as to identify the causal effects of policy interventions more precisely. Causal effects.

The transportation sector is characterized by high complexity and significant spatial heterogeneity. In the study of air transportation, there are significant differences in the policy implementation effect, the progress of resumption of work and production, and the socio-economic background in different regions. The double difference model is able to capture such regional differences by comparing the changes in the treatment and control groups before and after the implementation of the policy, thus revealing the heterogeneous effects of the policy in time and space. In addition, it has been shown that the double difference model has strong analytical power in the transportation domain, especially in analyzing the effects of policy or facility changes on specific economic or social variables, and exhibits high applicability and robustness.

This study aims to quantify the causal effects of epidemic prevention and control policies on air transportation. Based on the structural properties of panel data, the double difference model provides a powerful tool to reveal the spatio-temporal impact of the policy by simultaneously exploiting the dynamics before and after the implementation of the policy, as well as the differential response of different regions. Compared to other models, the double diff

---

## [Decision Letter · Decision Letter 1]

12 Feb 2025

PONE-D-24-36622R1Analysis of the temporal and spatial evolution of China's air passenger transportation and the impact of policies on passenger flowsPLOS ONE

Dear Dr. LUO,

Thank you for submitting your manuscript to PLOS ONE. After careful consideration, we feel that it has merit but does not fully meet PLOS ONE’s publication criteria as it currently stands. Therefore, we invite you to submit a revised version of the manuscript that addresses the points raised during the review process.

We look forward to receiving your revised manuscript.

Kind regards,

Qing-Chang Lu

Academic Editor

PLOS ONE

**Additional Editor Comments:**

The reviewers' are not satisfied with the revisions. The atuhors are suggested to revise and response to the comments of the reviewers seriously.

Reviewers' comments:

Reviewer's Responses to Questions

**Comments to the Author**

1. If the authors have adequately addressed your comments raised in a previous round of review and you feel that this manuscript is now acceptable for publication, you may indicate that here to bypass the “Comments to the Author” section, enter your conflict of interest statement in the “Confidential to Editor” section, and submit your "Accept" recommendation.

Reviewer #1: (No Response)

Reviewer #2: All comments have been addressed

2. Is the manuscript technically sound, and do the data support the conclusions?

Reviewer #1: (No Response)

Reviewer #2: Yes

3. Has the statistical analysis been performed appropriately and rigorously? 

Reviewer #1: (No Response)

Reviewer #2: Yes

4. Have the authors made all data underlying the findings in their manuscript fully available?

Reviewer #1: (No Response)

Reviewer #2: Yes

5. Is the manuscript presented in an intelligible fashion and written in standard English?

Reviewer #1: (No Response)

Reviewer #2: Yes

6. Review Comments to the Author

Reviewer #1: I dont think the authors have treated the revision serisouly. Again, please search and learn how other fine studies' structure and writing styles and present your wok in a good and appropriate way. Both Abstract, Introduction and Literature review still needs a big improvement (Sttructure, languages and the academic way how you present them). Also, in literature review, do not just copy and paste how other scholars done their works, summarize them, compare them and present it in your own words. Figure quality still not bad, improve the quality to meet the standard and let readers easy to catch up.

Reviewer #2: 1.The innovations of this paper should be more clearly highlighted.

2. In the author information section, should the same organization be listed repeatedly? Please carefully verify the journal's formatting requirements.

3. It is suggested to indicate the enlarged area in the original image with a rectangle.

4. In section 2.3.1, it is recommended to place the introduction of the parameters after the appearance of the formula, and the same approach should be applied to the other formulas.

7. PLOS authors have the option to publish the peer review history of their article (what does this mean? ). If published, this will include your full peer review and any attached files.

**Do you want your identity to be public for this peer review?** For information about this choice, including consent withdrawal, please see our Privacy Policy .

Reviewer #1: No

Reviewer #2: No

---

## [Author Response · Author response to Decision Letter 2]

26 Mar 2025

Response to Reviewer #1

Response:

We thank the referee very much for carefully reviewing our paper and providing the very nice and constructive suggestions, which help improve the quality of this paper even further. It is our pleasure. In this revision, we have followed the suggestions to revise the paper accordingly, with all the revisions highlighted with red color.

1. I dont think the authors have treated the revision serisouly. Again, please search and learn how other fine studies' structure and writing styles and present your wok in a good and appropriate way. Both Abstract, Introduction and Literature review still needs a big improvement (Sttructure, languages and the academic way how you present them). Also, in literature review, do not just copy and paste how other scholars done their works, summarize them, compare them and present it in your own words. Figure quality still not bad, improve the quality to meet the standard and let readers easy to catch up.

Response to 1

Thank you for your valuable feedback. We have carefully reviewed relevant high-quality studies to refine the structure, language, and academic presentation of our manuscript. In particular, we have made comprehensive revisions to the abstract, introduction, and literature review to improve clarity and coherence.Additionally, we have reorganized the literature review to provide a more structured summary and comparison of previous studies, ensuring that our discussion is presented in our own words to maintain academic rigor.We appreciate your positive comments on the quality of our figures and tables.

Response to Reviewer #2

Response:

We thank the referee very much for the feedback and nice suggestions. In this revision, we have followed the suggestions to revise the paper accordingly, with all the revisions highlighted with red color.

1.The innovations of this paper should be more clearly highlighted.

Response to 1

Thank you for your constructive suggestions. We have further emphasized the novelty of our study by explicitly outlining its unique contributions in the introduction (pages 6, lines 113–124).

2. In the author information section, should the same organization be listed repeatedly? Please carefully verify the journal's formatting requirements.

Response to 2

We appreciate your attention to detail regarding formatting. We have carefully checked the journal’s formatting guidelines and have made the necessary adjustments to the author information section to ensure compliance with the journal’s requirements.

3. It is suggested to indicate the enlarged area in the original image with a rectangle.

Response to 3

Following your recommendation, we have modified the figures by using rectangular markers to indicate magnified areas, thereby enhancing the clarity and readability of the visual representations.

4. In section 2.3.1, it is recommended to place the introduction of the parameters after the appearance of the formula, and the same approach should be applied to the other formulas.

Response to 4

We also appreciate your meticulous review of the formula arrangement. In Section 2.3.1, we have revised the order of presentation, placing parameter descriptions after the equations for better logical flow. Additionally, we have applied this adjustment consistently throughout the manuscript to enhance readability and coherence.

---

## [Decision Letter · Decision Letter 2]

31 Mar 2025

PONE-D-24-36622R2Analysis of the temporal and spatial evolution of China's air passenger transportation and the impact of policies on passenger flowsPLOS ONE

Dear Dr. LUO,

Thank you for submitting your manuscript to PLOS ONE. After careful consideration, we feel that it has merit but does not fully meet PLOS ONE’s publication criteria as it currently stands. Therefore, we invite you to submit a revised version of the manuscript that addresses the points raised during the review process.

We look forward to receiving your revised manuscript.

Kind regards,

Qing-Chang Lu

Academic Editor

PLOS ONE

Journal Requirements:

**Additional Editor Comments:**

The authors are suggested to revise the literature review section.

Reviewers' comments:

Reviewer's Responses to Questions

**Comments to the Author**

1. If the authors have adequately addressed your comments raised in a previous round of review and you feel that this manuscript is now acceptable for publication, you may indicate that here to bypass the “Comments to the Author” section, enter your conflict of interest statement in the “Confidential to Editor” section, and submit your "Accept" recommendation.

Reviewer #1: (No Response)

Reviewer #2: All comments have been addressed

2. Is the manuscript technically sound, and do the data support the conclusions?

Reviewer #1: (No Response)

Reviewer #2: Yes

3. Has the statistical analysis been performed appropriately and rigorously? 

Reviewer #1: (No Response)

Reviewer #2: Yes

4. Have the authors made all data underlying the findings in their manuscript fully available?

Reviewer #1: (No Response)

Reviewer #2: Yes

5. Is the manuscript presented in an intelligible fashion and written in standard English?

Reviewer #1: (No Response)

Reviewer #2: Yes

6. Review Comments to the Author

Reviewer #1: Writting a fine literature review, you need to use "your own" academic words to represent the literatrures you found related. Certainly, you can use one or two scholars' sentences, such as "A said...", "B used ... to do...". But you cannot use this structure everywhere in your literature review section.

Also, please make the legend in your figures more clearly to see.

Reviewer #2: The author has addressed the previously raised concerns with appropriate revisions. The research content shows some relevance, and therefore, I recommend this manuscript for publication.

7. PLOS authors have the option to publish the peer review history of their article (what does this mean? ). If published, this will include your full peer review and any attached files.

**Do you want your identity to be public for this peer review?** For information about this choice, including consent withdrawal, please see our Privacy Policy .

Reviewer #1: No

Reviewer #2: No

---

## [Author Response · Author response to Decision Letter 3]

7 Apr 2025

Academic Editor

1.Please review your reference list to ensure that it is complete and correct. If you have cited papers that have been retracted, please include the rationale for doing so in the manuscript text, or remove these references and replace them with relevant current references. Any changes to the reference list should be mentioned in the rebuttal letter that accompanies your revised manuscript. If you need to cite a retracted article, indicate the article’s retracted status in the References list and also include a citation and full reference for the retraction notice.

Response to 1

Thank you for your valuable comments regarding our reference list. We have conducted a thorough review of all citations to ensure their completeness and accuracy, and can confirm that no retracted papers have been included. All references have been carefully formatted according to the journal's guidelines, with each entry verified for proper presentation. Should any further modifications be required prior to final publication, please do not hesitate to let us know - we would be happy to make any necessary adjustments.

Response to Reviewer #1

Response:

We thank the referee very much for carefully reviewing our paper and providing the very nice and constructive suggestions, which help improve the quality of this paper even further. It is our pleasure. In this revision, we have followed the suggestions to revise the paper accordingly, with all the revisions highlighted with red color.

1.Writting a fine literature review, you need to use "your own" academic words to represent the literatrures you found related. Certainly, you can use one or two scholars' sentences, such as "A said...", "B used ... to do...". But you cannot use this structure everywhere in your literature review section.

Also, please make the legend in your figures more clearly to see.

Response to 1

We sincerely appreciate your constructive feedback on our manuscript. In response to your valuable suggestions, we have thoroughly revised the literature review section to present research findings in a more integrated manner while minimizing excessive direct citations. Additionally, we have enhanced all figure legends by incorporating them as high-resolution TIFF files to ensure optimal clarity when enlarged. We are grateful for your insightful comments and believe these revisions have significantly improved our manuscript. Should you have any further suggestions, we would be pleased to address them.

Response to Reviewer #2

Response:

We thank the referee very much for the feedback and nice suggestions.

1.The author has addressed the previously raised concerns with appropriate revisions. The research content shows some relevance, and therefore, I recommend this manuscript for publication.

Response to 1

We sincerely appreciate your time and valuable feedback throughout the review process. Thank you for recognizing our efforts in addressing the previous concerns and for your positive recommendation regarding our manuscript's publication. We are grateful for your constructive comments, which have significantly contributed to improving the quality of our work.

Should there be any additional minor revisions required prior to final publication, please do not hesitate to let us know. We would be happy to make any necessary adjustments.

Thank you again for your thoughtful review and support.

---

## [Editor Report · Decision Letter 3]

9 Apr 2025

Analysis of the temporal and spatial evolution of China's air passenger transportation and the impact of policies on passenger flows

PONE-D-24-36622R3

Dear Dr. LUO,

We’re pleased to inform you that your manuscript has been judged scientifically suitable for publication and will be formally accepted for publication once it meets all outstanding technical requirements.

Kind regards,

Qing-Chang Lu

Academic Editor

PLOS ONE

Additional Editor Comments (optional):

The authors have addressed the comments raised by the reviewers.
---

## [Editor Report · Acceptance letter]

PONE-D-24-36622R3

PLOS ONE

Dear Dr. Luo,

I'm pleased to inform you that your manuscript has been deemed suitable for publication in PLOS ONE. Congratulations! Your manuscript is now being handed over to our production team.

Kind regards,

on behalf of

Dr. Qing-Chang Lu

Academic Editor

PLOS ONE